# Effects of Compression Pants with Different Pressure Levels on Anaerobic Performance and Post-Exercise Physiological Recovery: Randomized Crossover Trial

**DOI:** 10.3390/s25154875

**Published:** 2025-08-07

**Authors:** Qinlong Li, Kaixuan Che, Wenlang Yu, Wenda Song, Yue Zhou

**Affiliations:** 1Department of Exercise Physiology, Exercise Science School, Beijing Sport University, Beijing 100084, China; liqinlong@bsu.edu.cn (Q.L.); 2023210312@bsu.edu.cn (W.S.); 2Department of Exercise Biochemistry, Exercise Science School, Beijing Sport University, Beijing 100084, China; 2019210241@bsu.edu.cn; 3Department of Physical Fitness and Health, Exercise Science School, Beijing Sport University, Beijing 100084, China; wlyu@bsu.edu.cn

**Keywords:** compression pants, lower limb muscles, anaerobic performance, post-exercise recovery

## Abstract

Compression pants, as functional sportswear providing external pressure, are widely used to enhance athletic performance and accelerate recovery. However, systematic investigations into their effectiveness during anaerobic exercise and the impact of different pressure levels on performance and post-exercise recovery remain limited. This randomized crossover controlled trial recruited 20 healthy male university students to compare the effects of four garment conditions: non-compressive pants (NCP), moderate-pressure compression pants (MCP), high-pressure compression pants (HCP), and ultra-high-pressure compression pants (UHCP). Anaerobic performance was assessed through vertical jump, agility tests, and the Wingate anaerobic test, with indicators including time at peak power (TPP), peak power (PP), average power (AP), minimum power (MP), power drop (PD), and total energy produced (TEP). Post-exercise blood lactate concentrations and heart rate responses were also monitored. The results showed that both HCP and UHCP significantly improved vertical jump height (*p* < 0.01), while MCP outperformed all other conditions in agility performance (*p* < 0.05). In the Wingate test, MCP achieved a shorter TPP compared to NCP (*p* < 0.05), with significantly higher AP, lower PD, and greater TEP than all other groups (*p* < 0.05), whereas HCP showed an advantage only in PP over NCP (*p* < 0.05). Post-exercise, all compression pant groups recorded significantly higher peak blood lactate (Lamax) levels than NCP (*p* < 0.05), with MCP showing the fastest lactate clearance rate. Heart rate analysis revealed that HCP and UHCP induced higher maximum heart rates (HR_max_) (*p* < 0.05), while MCP exhibited superior heart rate recovery at 3, 5, and 10 min post-exercise (*p*< 0.05). These findings suggest that compression pants with different pressure levels yield distinct effects on anaerobic performance and physiological recovery. Moderate-pressure compression pants demonstrated the most balanced and beneficial outcomes across multiple performance and recovery metrics, providing practical implications for the individualized design and application of compression garments in athletic training and rehabilitation.

## 1. Introduction

Enhancing athletic performance and improving post-exercise recovery efficiency are key topics in sports science research. With the continuous advancement of exercise physiology and training methodologies, in addition to traditional approaches such as training programs and nutritional supplementation, sports equipment—particularly functional apparel—has emerged as a potential aid in optimizing performance and accelerating physiological recovery. In recent years, compression garments (e.g., compression pants, sleeves, and tops) have gained increasing popularity in athletic and rehabilitation settings. These garments apply external pressure to the body, aiming to improve blood circulation, support muscle groups, reduce exercise-induced injuries, and facilitate recovery, and are now widely used among athletes, fitness enthusiasts, and rehabilitation populations [1,2,3,4].

Previous studies indicated that compression pants may positively influence both performance enhancement and post-exercise recovery; however, most research to date has primarily focused on their effects during aerobic exercise or the recovery phase following endurance activities [1]. Some researchers have suggested that tight-fitting garments can enhance endurance running performance by increasing maximal oxygen uptake (VO_2_max) [4] and improving running economy [5]. A systematic meta-analysis encompassing 42 studies and 769 participants found that compression garments significantly improved speed, endurance, and functional movement performance, with more pronounced effects observed in highly trained athletes. This study also highlighted the region-specific effectiveness of compression gear depending on the type of exercise: for instance, lower-limb compression was more beneficial for enhancing endurance performance, whereas full-body compression was more effective in improving speed [6]. In a randomized crossover trial, Kim et al. (2021) demonstrated that wearing a full-body compression suit significantly enhanced VO_2_max, exercise duration, and anaerobic threshold in recreational male marathon runners, as well as improving minimum power output and 3 km run performance compared to non-compression conditions [7]. These findings suggest that compression garments may simultaneously enhance both aerobic and anaerobic capacities, thereby optimizing overall physical performance [7]. Further investigations have indicated that these benefits are highly dependent on the “interface pressure” between the compression pants and the skin. In a single-blind crossover study, Williams et al. (2020) confirmed that high-compression (HC) garments significantly improved 8 km cycling time-trial performance and accelerated blood lactate clearance following consecutive days of high-intensity cycling, suggesting a modulatory role of pressure levels in performance maintenance [8]. Similarly, McManus (2020) found that compression pants effectively improved energy efficiency during running, as evidenced by enhanced running economy, further supporting their potential for aerobic performance enhancement [9].

Compared with standard athletic pants, compression pants have been shown to enhance proprioception and muscle function during post-exercise fatigue [10,11]. They also help reduce delayed onset muscle soreness (DOMS) during passive recovery by promoting hemodynamic responses, thereby improving subsequent performance [12,13,14]. A recent study demonstrated that, compared to standard athletic pants, wearing compression pants during a 60 min passive recovery period significantly increased stroke volume and cardiac output, while reducing heart rate and blood lactate concentration. Furthermore, in a subsequent 5 min maximal cycling test, participants wearing compression pants exhibited improved power output and pedaling cadence, suggesting that compression garments can enhance recovery quality and subsequent performance by optimizing blood flow responses and reducing perceived fatigue [15]. Research utilizing near-infrared spectroscopy (NIRS) has further shown that within a specific pressure range (approximately 15–30 mmHg), compression pants can effectively improve tissue oxygen saturation, oxyhemoglobin, and deoxyhemoglobin levels during rest and low-intensity activity, thereby modulating microcirculatory status and providing a physiological basis for recovery [16]. Furthermore, several studies have confirmed that compression garments may accelerate muscle strength recovery, enhance lactate clearance, and regulate creatine kinase (CK) activity during post-exercise recovery, suggesting their potential benefits in mitigating exercise-induced muscle damage and supporting metabolic recovery [2,11,13,14]. A systematic review encompassing 160 studies and 2530 participants further supports this perspective, indicating that the use of compression garments during the post-exercise recovery phase is closely associated with reductions in lactate dehydrogenase (LDH) levels and consistently demonstrates significant alleviation of perceived muscle soreness. These findings highlight the positive role of compression garments in fatigue attenuation and metabolic recovery [17].

However, research into the effects and mechanisms of compression pants during anaerobic exercise remains limited. In particular, there is a lack of systematic studies exploring how varying pressure levels of compression pants influence high-intensity anaerobic activities (e.g., sprinting, explosive jumping, resistance training) in healthy populations [3]. Preliminary findings suggest that compression pants may contribute to anaerobic performance by reducing muscle oscillation, enhancing muscle oxygenation, improving neuromuscular coordination, and delaying fatigue onset [1]; yet it remains unclear whether these benefits follow a pressure-dependent dose–response relationship [18]. Moreover, most existing studies have examined only a single model or pressure level of compression garments, lacking horizontal comparisons across a pressure gradient [3], which limits the generalizability of their findings. A few studies have attempted to evaluate the specific effects of compression pants on short-term high-intensity performance. For instance, one study involving 24 physically active men and women reported that, compared to standard athletic pants, compression pants provided slight improvements in 10 m sprint and change-of-direction tasks. However, these effects were mostly of small effect size and showed no significant differences in jump and balance tests, suggesting that the observed benefits might fall within the margin of measurement error [19]. This indicates that while compression pants may exert some influence on explosive or agility-related tasks, their actual effectiveness remains debatable.

It is worth noting that most current studies have primarily focused on endurance-trained male participants or mixed-gender samples, while research specifically targeting healthy, recreationally active males remains relatively limited. However, this population constitutes a high-frequency user group of compression pants in real-world fitness settings, particularly during anaerobic activities such as resistance training, interval workouts, and team-based sports. Given their widespread engagement in such high-intensity exercise modalities, understanding this group’s response characteristics to varying levels of compression pressure holds significant practical relevance and research value. Accordingly, the present study aims to examine the effects of compression pants across varying pressure levels on anaerobic performance in healthy recreationally active males. By comparing multiple pressure conditions, this study seeks to clarify the differential impacts of compression pressure on performance outcomes, thereby providing empirical evidence and theoretical support for the personalized design and practical application of compression garments in sports and exercise settings.

## 2. Participants and Methods

### 2.1. Participants

Participants were recruited through on-campus posters. Inclusion criteria were as follows: healthy male individuals aged 18 to 25 years, engaging in regular physical activity (≥3 sessions per week), but without systematic athletic training experience. To eliminate potential interference from injuries, participants were excluded if they had any lower-limb musculoskeletal injuries within the past six months. To ensure standardization of compression pant sizing and improve consistency and comparability of the intervention, participants were required to have a height between 175 and 180 cm and a body weight between 60 and 70 kg, allowing all subjects to wear the same garment size. Based on a priori power analysis referencing previous studies [15], an effect size of 0.3, significance level α = 0.05, and statistical power of 1 − β = 0.80 were used, yielding a minimum required sample size of 20. Ultimately, 20 eligible participants were successfully recruited and completed all testing procedures, and their data were included in the final analysis (Table 1).

This study was conducted in accordance with the principles of the Declaration of Helsinki and was approved by the Ethics Committee of Beijing Sport University (Approval No. 2024160H). All participants provided written informed consent before this study began and were fully informed of the study procedures and potential risks.

### 2.2. Experimental Design

This study adopted a randomized crossover controlled trial design. Each participant completed the full test protocol under four different garment conditions: non-compressive pants (NCP), moderate-pressure compression pants (MCP), high-pressure compression pants (HCP), and ultra-high-pressure compression pants (UHCP) (Figure 1A). A one-week washout period was scheduled between each condition to minimize training or fatigue-related interference. To ensure data consistency, all tests were conducted between 14:00 and 16:00. Participants were instructed to refrain from consuming stimulants prior to testing and to avoid strenuous physical activity on the test day. Upon arrival at the laboratory, participants first underwent body composition assessment, followed by the collection of resting heart rate and baseline blood lactate concentrations. Participants then wore the assigned compression pants for the session. After completing a standardized warm-up, the pressure values exerted by the three compression pants (MCP, HCP, UHCP) on the rectus femoris, vastus lateralis, gastrocnemius, and tibialis anterior muscles in the standing position were measured. Subsequently, participants completed three performance tests in sequence: the Illinois agility test, a vertical jump test, and a 30 s Wingate anaerobic test. A 15 min passive rest interval was provided between each test to reduce fatigue effects on subsequent performance. Immediately following the Wingate test, heart rate and blood lactate concentration were recorded at 0, 3, 5, and 10 min post-exercise to evaluate physiological recovery (Figure 1B).

### 2.3. Methods

#### 2.3.1. Compression Pressure Assessment

Three commercially available compression tights were selected for the experiment: ZuoYouLiLiang (ZuoYouLiLiang Sportswear Co., Fujian, China), CW-X (Wacoal Corp., Kyoto, Japan), and SKINS (SKINS International Trading AG, Steinhausen, Switzerland). Garment pressure was measured using a Texsens clothing pressure tester (novel GmbH, Munich, Germany), a flexible thin-film sensor system designed for measuring low contact pressures at the interface between the human body and textile surfaces. The device assessed the three compression-pant models in the upright position, with all garments sized to accommodate individuals 175–180 cm in height and 60–70 kg in body mass. The Texsens system has a measurement range of 1–10 kPa, a sampling frequency of 50 Hz, and a circular sensor area with a diameter of 10 mm. Data were exported in ASCII format for processing. The sensor is highly conformable, with minimal structural interference, enabling stable static pressure measurements even under mild skin deformation and temperature variation. Pressure values were ultimately expressed in mmHg. During testing, each participant stood erect while sensors were placed sequentially over the rectus femoris, vastus lateralis, gastrocnemius, and tibialis anterior (Figure 2). Three readings were taken at each site and averaged to enhance measurement reliability and accuracy. Based on the measured pressure values at rest, these garments were categorized into three compression levels: MCP, HCP, and UHCP, respectively [20].

#### 2.3.2. Vertical Jump Test

The vertical jump test was used to assess lower-limb explosive power. A vertical jump mat (ZT-II, Beijing Xindong Huateng Technology Co., Ltd., Beijing, China) was employed to measure jump height during a countermovement jump. Participants were instructed to place their hands on their hips and perform a rapid downward squat followed by a maximal vertical jump using a standardized technique.

#### 2.3.3. Illinois Agility Test

The Illinois Agility Test (IAT) was used to assess participants’ agility, explosive power, and coordination. The testing protocol followed established procedures described in previous studies [21]. The test was conducted on a flat, even surface, with cones used to mark the start, finish, and turning points. A stopwatch was used to record completion time. Shorter completion times indicated higher levels of agility and explosive performance. This test is particularly relevant for sports that require frequent changes in direction and rapid responses.

#### 2.3.4. 30-Second Wingate Test

Anaerobic performance was assessed using the 30 s Wingate test on a cycle ergometer (Monark 894E, Monark Exercise AB, Vansbro, Sweden). The resistance load was set at 0.075 × body weight (kg), following standardized procedures [22]. Participants began pedaling from a stationary position and accelerated to their maximum cadence in a seated posture. Once peak speed was reached, the predetermined brake weight was rapidly applied, and participants continued pedaling at maximal effort for 30 s. Verbal encouragement was provided throughout the test to ensure maximum exertion. Performance metrics recorded during the test included brake weight (BW), time at peak power (TPP), peak power (PP), average power (AP), minimum power (MP), power drop (PD), and total energy produced (TEP).

#### 2.3.5. Blood Lactate Measurement

Capillary blood samples were collected from the fingertip at rest and at 0, 3, 5, and 10 min following the Wingate anaerobic test. Approximately 20 µL of blood was drawn at each time point using a sterile, single-use lancet. Blood lactate concentrations were analyzed immediately using a portable lactate analyzer (EKF Biosen, Barleben, Germany).

Post-exercise lactate clearance rate was calculated to evaluate metabolic recovery. Specifically, V(10 min) represented the lactate clearance rate between 3 and 10 min post-exercise, calculated using the blood lactate concentrations at 3 min [La(3 min)] and 10 min [La(10 min)], and their corresponding time points, T(3 min) and T(10 min), respectively.(1)$$V(10\;min)=\frac{La\left(3\;min\right)-La\left(10\;min\right)}{T\left(10\;min\right)-T\left(3\;min\right)}$$

#### 2.3.6. Heart Rate Monitoring

Heart rate was continuously monitored using a Polar H10 heart rate chest strap (Polar Electro, Kempele, Finland). Participants wore the chest strap starting from the resting state, and heart rate was recorded continuously from pre-test through to 10 min post-30 s Wingate test. Resting heart rate and post-exercise heart rate values at 0, 3, 5, and 10 min were collected and recorded. The 0 min value reflects the maximum heart rate response immediately after exercise; the 3 and 5 min values represent the early recovery phase, characterized by rapid parasympathetic reactivation, and the 10 min point reflects the later stage of recovery when heart rate begins to stabilize. These time points were consistent with those used for blood lactate sampling, allowing for a synchronized assessment of cardiovascular and metabolic recovery. Heart rate recovery (HRR) was calculated to evaluate post-exercise autonomic recovery. This approach provides a comprehensive assessment of autonomic regulatory dynamics and is widely regarded as a reliable and non-invasive method for evaluating post-exercise recovery capacity [23,24,25]. Specifically, R(n min) represents the heart rate recovery rate at n minutes post-exercise, calculated based on HR at 0 min [HR(0 min)], HR at n minutes [HR(n min)], and resting heart rate [HR(rest)].(2)$$R\left(n\;min\right)=\frac{HR\left(0\;min\right)-HR\left(n\;min\right)}{HR\left(0\;min\right)-HR\left(rest\right)}\times 100\%$$

### 2.4. Statistical Analysis

All statistical analyses were performed using SPSS software (version 26.0, IBM Corp., Armonk, NY, USA). The data were presented as the mean ± standard deviation ($\bar{x}$ ± s). The Shapiro–Wilk test was first used to assess the normality of the data distribution. One-way repeated measures analysis of variance (ANOVA) was conducted to compare the effects of different compression pant pressure levels on performance and recovery outcomes. When significant differences were identified, Bonferroni post hoc tests were applied for pairwise comparisons. To account for the crossover design, potential period and sequence (carryover) effects were additionally examined using a linear mixed-effects model, in which “condition” and “period” were treated as fixed effects, and “subject” was included as a random effect. No significant carryover effects were found. A *p*-value of <0.05 was considered statistically significant.

## 3. Results

### 3.1. Interface Pressure Values on Lower-Limb Muscle Regions Under Different Compression Pants

The rectus femoris and vastus lateralis are key components of the quadriceps femoris, primarily responsible for knee extension and coordinated hip flexion. These muscles serve as the main drivers of explosive force output and peak power generation during typical anaerobic activities such as jumping and sprinting. The gastrocnemius, as the primary plantar flexor, plays a central role in lower-limb propulsion and the conversion of ground reaction forces, making it essential in high-intensity movements such as jumping, take-off, and directional changes. Although not a primary force-producing muscle, the tibialis anterior controls ankle dorsiflexion and plays a critical role in ankle joint stabilization, landing impact absorption, and gait adjustment during rapid movement, thereby contributing significantly to agility and coordination. These four muscles were selected to assess the distribution of interface pressure applied by compression garments at key lower-limb regions, providing a representative profile across functionally important muscle groups.

As shown in Figure 3, the interface pressure values measured on four lower-limb muscle groups (rectus femoris, vastus lateralis, tibialis anterior, and lateral gastrocnemius) varied significantly among the three compression pant conditions: moderate-pressure (MCP), high-pressure (HCP), and ultra-high-pressure (UHCP).

For the rectus femoris, the UHCP group exhibited significantly higher pressure compared to both the MCP group (30.2 ± 6.49 mmHg vs. 21.11 ± 4.41 mmHg, *p* = 0.000) and the HCP group (30.2 ± 6.49 mmHg vs. 25.75 ± 6.57 mmHg, *p* = 0.021). The HCP group also showed significantly higher pressure than the MCP group (*p* = 0.016).

For the vastus lateralis, pressure in the UHCP group was significantly higher than that in both the MCP group (29.51 ± 5.21 mmHg vs. 21.8 ± 4.97 mmHg, *p* = 0.000) and the HCP group (29.51 ± 5.21 mmHg vs. 26.15 ± 4.61 mmHg, *p* = 0.035), while that of the HCP group was also significantly higher than that of the MCP group (*p* = 0.007).

For the tibialis anterior, the UHCP group again exhibited the highest pressure, significantly exceeding both the MCP (29.51 ± 3.52 mmHg vs. 21.87 ± 4.31 mmHg, *p* = 0.000) and HCP groups (29.51 ± 3.52 mmHg vs. 24.28 ± 3.36 mmHg, *p* = 0.000). The HCP group also applied significantly more pressure than the MCP group (*p* = 0.047).

For the lateral gastrocnemius, the UHCP group showed the highest pressure, significantly greater than that of the MCP group (29.51 ± 3.53 mmHg vs. 23.89 ± 5.24 mmHg, *p* = 0.001), and the HCP group also demonstrated significantly higher pressure than the MCP group (27.75 ± 6.39 mmHg vs. 23.89 ± 5.24 mmHg, *p* = 0.022).

### 3.2. Vertical Jump Height Under Different Compression Pant Pressure Levels

The vertical jump test primarily reflects the explosive strength of the lower-limb muscles, particularly the synergistic power generation of the quadriceps femoris (including the rectus femoris and vastus lateralis) and the gastrocnemius. This test is highly representative and is commonly used to assess performance in strength-based and short-duration high-intensity sports. To investigate how different levels of compression affect explosive performance, we compared vertical jump outcomes under various compression conditions. As shown in Figure 4, participants wearing high-pressure compression pants (HCP) achieved significantly greater vertical jump height compared to those wearing moderate-pressure compression pants (MCP) (52.01 ± 2.86 cm vs. 48.97 ± 1.62 cm, *p* = 0.001) and non-compressive pants (NCP) (52.01 ± 2.86 cm vs. 47.63 ± 3.47 cm, *p* = 0.000). Additionally, the ultra-high-pressure compression pants (UHCP) group also exhibited significantly higher jump height than the NCP group (50.41 ± 2.49 cm vs. 47.63 ± 3.47 cm, *p* = 0.000). However, no significant difference was found between UHCP and MCP (50.41 ± 2.49 cm vs. 48.97 ± 1.62 cm, *p* = 0.094).

### 3.3. Agility Performance Under Different Compression Pant Pressure Levels

The Illinois agility test is a standardized and reliable assessment used to evaluate rapid change-of-direction ability, short-duration acceleration and deceleration, and neuromuscular coordination. Through high-frequency sprinting, abrupt stops, and sharp directional changes, it strongly activates the major lower-limb muscle groups—particularly the quadriceps femoris, gastrocnemius, and tibialis anterior—assessing their explosive force and control during both eccentric and concentric contractions. This test reflects the comprehensive physical demands of anaerobic activities on speed, agility, and lower-limb stability. To investigate the effects of different compression levels on agility performance, participants’ results were compared across various garment conditions. As shown in Figure 5, participants demonstrated significantly slower agility performance when wearing ultra-high-pressure compression pants (UHCP) (14.11 ± 0.97 s vs. 13.48 ± 0.79 s, *p* = 0.006), high-pressure compression pants (HCP) (13.98 ± 0.50 s vs. 13.48 ± 0.79 s, *p* = 0.027), and non-compressive pants (NCP) (14.24 ± 0.46 s vs. 13.48 ± 0.79 s, *p* = 0.001) compared to moderate-pressure compression pants (MCP).

### 3.4. 30-Second Wingate Test Results Under Different Compression Pant Pressure Levels

The 30 s Wingate test is considered the gold standard for evaluating peak anaerobic power, mean anaerobic power, and anaerobic capacity (fatigue index). It requires participants to perform an all-out sprint on a cycle ergometer for 30 s, fully engaging the anterior thigh and lower leg muscles, particularly the rectus femoris and vastus lateralis. This classic anaerobic assessment provides a reliable measure of short-duration maximal output capacity and fatigue development. To evaluate the impact of different compression levels on anaerobic performance, results from the 30 s Wingate test were analyzed across the four compression pant conditions (Table 2). For time to peak power (TPP), the MCP group achieved significantly shorter times compared to the NCP group (3778.13 ± 1142.47 ms vs. 4906.18 ± 1817.55 ms, *p* = 0.029). Regarding peak power (PP), the HCP group exhibited significantly higher values than the NCP group (10.51 ± 0.44 W/kg vs. 10.17 ± 0.29 W/kg, *p* = 0.014). For average power (AP), the MCP group demonstrated significantly higher output compared to the NCP (8.72 ± 0.89 W/kg vs. 8.09 ± 0.16 W/kg, *p* = 0.000), HCP (8.72 ± 0.89 W/kg vs. 8.07 ± 0.26 W/kg, *p* = 0.000), and UHCP groups (8.72 ± 0.89 W/kg vs. 8.07 ± 0.19 W/kg, *p* = 0.000). For minimum power (MP), no significant differences were observed among the four groups. In terms of power drop (PD), the MCP group showed significantly lower values than the NCP group (0.14 ± 0.02 W/s/kg vs. 0.16 ± 0.02 W/s/kg, *p* = 0.031), HCP (0.14 ± 0.02 W/s/kg vs. 0.17 ± 0.03 W/s/kg, *p* = 0.008), and UHCP (0.14 ± 0.02 W/s/kg vs. 0.17 ± 0.04 W/s/kg, *p* = 0.011). For total energy produced (TEP), the MCP group generated significantly more energy than the NCP (15,038.57 ± 891.72 J vs. 14,430.77 ± 631.59 J, *p* = 0.020), HCP (15,038.57 ± 891.72 J vs. 14,486.36 ± 691.59 J, *p* = 0.034), and UHCP (15,038.57 ± 891.72 J vs. 14,393.38 ± 971.91 J, *p* = 0.014) groups.

### 3.5. Post-Exercise Blood Lactate Responses Under Different Compression Pant Pressure Levels

As shown in Figure 6A, following the 30 s Wingate test, the peak blood lactate concentration (LAmax) was significantly higher in the UHCP (17.24 ± 2.10 mmol/L vs. 15.47 ± 1.36 mmol/L, *p* = 0.003), HCP (16.87 ± 2.27 mmol/L vs. 15.47 ± 1.36 mmol/L, *p* = 0.017), and MCP (16.77 ± 1.29 mmol/L vs. 15.47 ± 1.36 mmol/L, *p* = 0.027) groups compared to the NCP group. As illustrated in Figure 6B, the lactate clearance rate at 10 min post-exercise was significantly higher in the MCP group than in the NCP group (0.36 ± 0.35 mmol/L vs. 0.06 ± 0.30 mmol/L, *p* = 0.007). Figure 6C presents the temporal trends of blood lactate concentrations before and after exercise. All groups showed a rapid rise in lactate levels immediately post-exercise, peaking at approximately 3 min, followed by a gradual decline over time.

### 3.6. Post-Exercise Heart Rate Responses Under Different Compression Pant Pressure Levels

As shown in Figure 7A, the HCP group exhibited significantly higher maximum heart rate (HRmax) compared to the NCP (188.06 ± 4.03 bpm vs. 182.25 ± 2.25 bpm, *p* = 0.000), MCP (188.06 ± 4.03 bpm vs. 183.75 ± 3.5 bpm, *p* = 0.000), and UHCP (188.06 ± 4.03 bpm vs. 185.83 ± 1.45 bpm, *p* = 0.021) groups. In addition, the UHCP group showed significantly higher HRmax than both the NCP (185.83 ± 1.45 bpm vs. 182.25 ± 2.25 bpm, *p* = 0.000) and MCP (185.83 ± 1.45 bpm vs. 183.75 ± 3.5 bpm, *p* = 0.031) groups. As illustrated in Figure 7B, no significant differences were observed among the groups in average heart rate (HRavg) during the test (*p* > 0.05). Regarding heart rate recovery (HRR) (Figure 7C), the MCP group demonstrated significantly better recovery at 3, 5, and 10 min post-exercise compared to the NCP group (3 min: 35.17 ± 6.24% vs. 28.86 ± 7.36%, *p* = 0.002; 5 min: 47.94 ± 2.91% vs. 39.34 ± 7.46%, *p* = 0.000; 10 min: 54.61 ± 5.69% vs. 50.21 ± 5.73%, *p* = 0.009), the HCP group (3 min: 35.17 ± 6.24% vs. 29.43 ± 7.46%, *p* = 0.005; 5 min: 47.94 ± 2.91% vs. 43.59 ± 3.56%, *p* = 0.003; 10 min: 54.61 ± 5.69% vs. 50.43 ± 6.44%, *p* = 0.014), and the UHCP group at 3 and 5 min (3 min: 35.17 ± 6.24% vs. 27.43 ± 3.21%, *p* = 0.000; 5 min: 47.94 ± 2.91% vs. 45.06 ± 1.91%, *p* = 0.046). Additionally, both the HCP (43.59 ± 3.56% vs. 39.34 ± 7.46%, *p* = 0.004) and UHCP (45.06 ± 1.91% vs. 39.34 ± 7.46%, *p* = 0.000) groups showed significantly higher heart rate recovery at 5 min compared to the NCP group. Figure 7D further illustrates the heart rate trajectories at rest and during the recovery period. All compression pant conditions showed faster post-exercise heart rate decline compared to NCP, with the most pronounced effect observed in the MCP group.

## 4. Discussion

This study systematically evaluated the effects of compression pants with three different pressure gradients on muscle interface pressure and anaerobic performance. The results revealed that compression pants significantly and selectively enhanced various aspects of anaerobic capacity, and these effects exhibited a clear pressure-dependent pattern. As shown in Figure 3, both ultra-high-pressure and high-pressure compression pants produced significantly greater interface pressure on key lower-limb muscle groups—namely the rectus femoris, vastus lateralis, tibialis anterior, and lateral gastrocnemius—compared to moderate-pressure compression pants (*p* < 0.05). With increasing garment pressure, the interface pressure values across all four muscle regions showed a progressive upward trend, confirming the graded pressure distribution effect exerted by the compression garments on different muscle groups.

This study revealed that compression pants exert pressure-dependent effects on athletic performance, characterized by task-specific and nonlinear patterns. While performance improvements were observed under multiple pressure conditions, the relationship between compression intensity and outcome did not follow a simple monotonic trend. Specifically, both high-pressure and ultra-high-pressure compression pants resulted in significantly greater vertical jump height compared to non-compressive pants (Figure 4), and the HCP group also exhibited significantly higher peak power (PP) than the NCP group (Table 2). These results suggest that in tasks requiring high explosive output, elevated external pressure may enhance performance by increasing muscle stiffness and tendon elasticity, thereby improving the efficiency of elastic energy storage and release [3]. Furthermore, based on rigid body dynamics theory, higher compression may help reduce muscle oscillation and energy dissipation, improving force transmission efficiency [11]. However, in agility testing (Figure 5) and the 30 s Wingate test (Table 2), the moderate-pressure compression pants (MCP) showed superior performance in sustained output, characterized by shorter TPP, reduced PD, and significantly higher AP and TEP. This indicates that an optimal level of compression may better support neuromuscular coordination and metabolic stability during complex or prolonged tasks. Compared to the HCP group, the superior agility performance observed with MCP may be attributed to enhanced sensorimotor integration. According to the sensory feedback modulation model, excessive external pressure may activate Golgi tendon organs, leading to the inhibition of α-motoneurons, reduced muscle recruitment, and impaired multi-joint coordination and reaction speed [26,27]. Moreover, while both HCP and UHCP showed explosive advantages in vertical jumping, they underperformed in sustained anaerobic power tasks, as reflected in significantly higher PD and lower TEP compared to the MCP group (*p* < 0.05; Table 2). This may be related to microvascular compression and reduced tissue oxygen saturation under high pressure [28], resulting in a rapid accumulation of hydrogen ions and anaerobic metabolites, thereby accelerating fatigue onset [29]. Collectively, these findings suggest that high-pressure compression pants may be more suitable for short-duration, high-intensity explosive tasks, whereas moderate-pressure pants are more advantageous for sustained output and change-of-direction performance. Rather than exhibiting a fixed dose–response trend, the performance outcomes varied with different pressure levels depending on the specific neuromuscular and physiological demands of the task. For instance, in the vertical jump test, performance improvements were observed as follows: HCP (+9.20%), UHCP (+5.84%), and MCP (+2.81%) relative to NCP. However, in tasks requiring quick directional changes or sustained power output, MCP showed the most favorable outcomes. This variable response pattern suggests that optimal compression may vary by exercise type, likely due to a trade-off among mechanical support, neuromuscular control, and local hemodynamics. Excessive pressure may impair blood perfusion and disrupt proprioceptive signaling, thereby limiting coordination and sustained muscular output [30]. A plausible explanation for these differential effects lies in the modulation of afferent feedback and motor command interaction, as described in the sensorimotor integration framework [31]. Moderate compression may enhance proprioceptive acuity and stabilize motor output without overwhelming sensory pathways, thereby improving coordination and efficiency in tasks requiring agility or endurance. In contrast, excessive pressure may saturate sensory input and impair central processing, leading to suboptimal motor control. This mechanism may help explain the superior performance observed with moderate-pressure garments in sustained or multidirectional tasks, despite smaller improvements in explosive output.

Blood lactate and heart rate are key physiological indicators reflecting metabolic stress and cardiovascular load during exercise, and are therefore crucial for evaluating the effects of compression garments on post-anaerobic recovery. In this study, peak blood lactate concentrations (LAmax) were elevated under all three compression conditions compared to the non-compressive pants (NCP), with the most pronounced increase observed in the ultra-high-pressure compression pants (UHCP) group (*p* < 0.01). This may be attributed to the restrictive effect of tight-fitting garments on local blood flow in the lower-limbs during exercise, potentially reducing skeletal muscle perfusion and oxygen availability [16], thereby increasing reliance on anaerobic metabolism and leading to greater lactate accumulation. Previous studies have shown that wearing lower-limb compression garments can accelerate the transfer of lactate from muscle tissue into the bloodstream and elevate blood lactate peaks—particularly under higher-pressure conditions. This effect is likely due to the external compression reducing local oxygen supply, thereby shifting muscle metabolism toward anaerobic glycolysis and lactate production [32,33]. Despite the higher lactate peaks observed in the high- and ultra-high-pressure groups, the moderate-pressure compression pants (MCP) group demonstrated significantly greater lactate clearance rates at all post-exercise time points. This suggests that moderate pressure may play a positive role in facilitating metabolite transport and clearance. It is possible that moderate compression promotes venous return in the lower limbs and reduces muscle oscillation [34,35,36], thereby accelerating the removal of lactate from muscle into circulation for subsequent aerobic metabolism. Studies have also shown that a certain level of compression in the lower limbs can improve peripheral circulation and enhance venous return [37], contributing to faster removal of metabolic byproducts and promoting physiological recovery after exercise [34]. The rate at which lactate is cleared post-exercise is often considered a reflection of aerobic metabolic capacity—faster clearance indicates more efficient oxidative metabolism and thus quicker recovery from metabolite-induced fatigue [38]. Therefore, the superior performance of moderate-pressure compression pants during recovery may be closely related to their beneficial effects on hemodynamics and metabolic homeostasis [39,40]. Heart rate recovery (HRR), an important indicator of autonomic nervous system reactivation, showed a similar pattern. The MCP group exhibited significantly greater heart rate recovery at 3, 5, and 10 min post-exercise compared to the other groups, suggesting that moderate pressure more effectively promotes early parasympathetic reactivation and stabilization of heart rate. A faster heart rate decline reflects timely vagal activation and more efficient cardiopulmonary regulation [41,42]. Thus, an optimal level of compression may enhance the muscle pump effect and sensory input, thereby upregulating vagal tone and accelerating heart rate recovery. In contrast, although high- and ultra-high-pressure garments induced higher peak heart rates (HRmax), they did not demonstrate corresponding recovery benefits, indicating that excessive compression may impose additional cardiovascular strain during early recovery.

In summary, compression pants with different pressure levels demonstrated task-dependent effects on athletic performance. Among them, MCP exhibited the most balanced overall performance, showing clear advantages in sustained power output and agility-based tasks, as well as in promoting post-exercise homeostatic recovery. While high-pressure and ultra-high-pressure compression pants showed benefits in enhancing short-duration explosive performance—such as vertical jump height and peak power—their excessive pressure levels were associated with increased local discomfort, restricted blood perfusion, and diminished power sustainability. These factors may limit their applicability in tasks requiring prolonged anaerobic output or multidirectional agility, and potentially interfere with metabolic efficiency during recovery. In contrast, MCP not only provides effective muscular support and proprioceptive feedback, but also demonstrates superior performance in output stability, reactive agility, and recovery. This makes it particularly suitable for sports that demand neuromuscular coordination, rapid directional changes, and endurance maintenance—such as basketball, volleyball, soccer, tennis, and long-distance running. Therefore, the selection of compression pressure should be optimized according to the specific demands of the athletic task, in order to enhance performance while maintaining metabolic balance and recovery efficiency.

Despite these findings, several limitations within the current investigation should be acknowledged, which also provide direction for future research in this area. The participant sample was relatively homogeneous, consisting only of young adult males with similar anthropometric characteristics, which limits the generalizability of the findings to broader populations such as females, older adults, elite athletes, or individuals with different body types. Future studies should include more diverse participant groups to improve external validity. In addition, due to the technical limitations of pressure sensors—namely instability and considerable measurement error during movement—we were unable to record reliable compression pressure during exercise; only static pressure data in resting conditions were collected. This constraint may limit the interpretation of the garments’ actual mechanical influence during high-intensity activities. Future work should consider incorporating motion-tolerant, real-time pressure monitoring systems. Furthermore, the present study did not include additional physiological or neuromuscular indicators such as electromyography (EMG), muscle oxygen saturation, or muscle stiffness, which could help elucidate the mechanisms behind the observed effects. Finally, this study focused only on acute responses to compression garments; the potential long-term training adaptation effects of repeated compression use during anaerobic training remain unknown. Future research should investigate the effects of compression garments across different populations, recruit larger sample sizes to support exploratory subgroup or individual-level analyses, explore chronic responses to compression during training, and incorporate additional physiological markers to comprehensively understand the multifaceted impact of compression on performance and recovery.

## 5. Conclusions

This study demonstrates that compression pants with different pressure levels have distinct effects on anaerobic performance and post-exercise recovery. High-pressure and ultra-high-pressure compression pants were effective in enhancing explosive vertical jump performance, making them more suitable for short-duration, high-intensity tasks. In contrast, moderate-pressure compression pants showed superior results in agility, average power, power maintenance, lactate clearance, and heart rate recovery, indicating their suitability for complex or sustained exercise scenarios. Overall, these findings support the importance of matching compression pressure levels to the specific demands of the sport and recovery needs. In particular, the balanced benefits of moderate-pressure compression pants highlight their value in both performance enhancement and recovery facilitation.

## Figures and Tables

**Figure 1 sensors-25-04875-f001:**
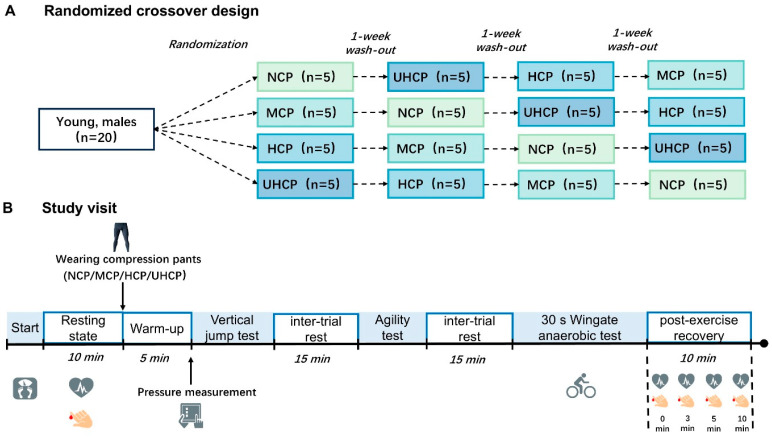
(**A**) Randomized crossover design. (**B**) Study visit. Participants completed four randomized crossover trials under different compression garment conditions (NCP, MCP, HCP, UHCP). Each session included body composition assessment, resting measurements, pressure testing, performance tests (Illinois agility test, vertical jump, and 30 s Wingate test), and post-exercise recovery monitoring at 0, 3, 5, and 10 min.

**Figure 2 sensors-25-04875-f002:**
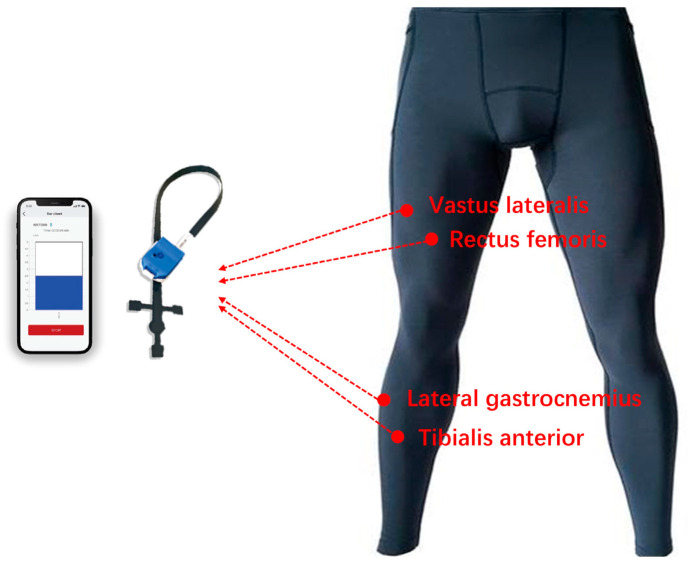
Compression garment pressure testing and sensor placement on lower-limb muscles.

**Figure 3 sensors-25-04875-f003:**
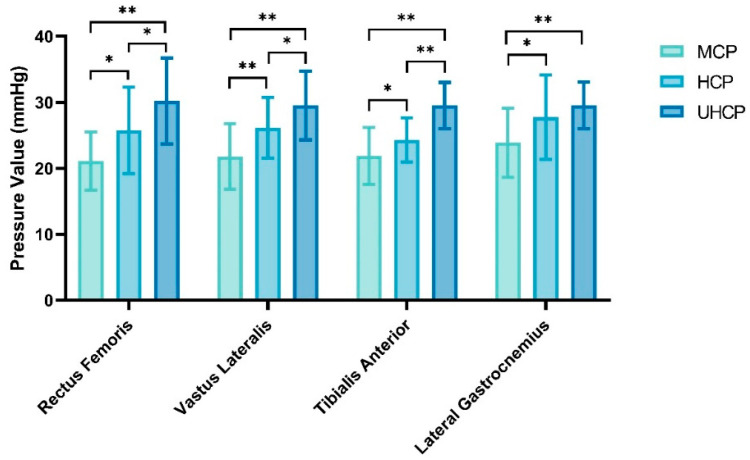
Interface pressure values at different lower-limb regions under compression pants of varying pressure levels (* *p* < 0.05, ** *p* < 0.01).

**Figure 4 sensors-25-04875-f004:**
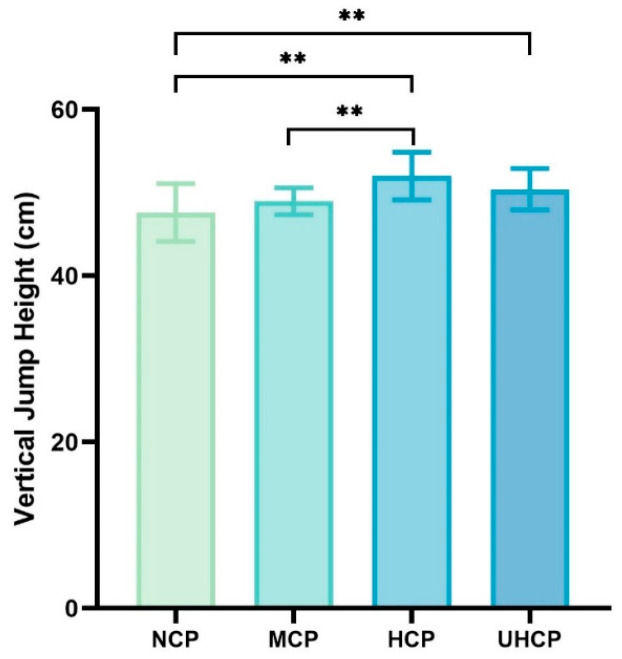
Vertical jump height under different compression pant pressure levels (** *p* < 0.01).

**Figure 5 sensors-25-04875-f005:**
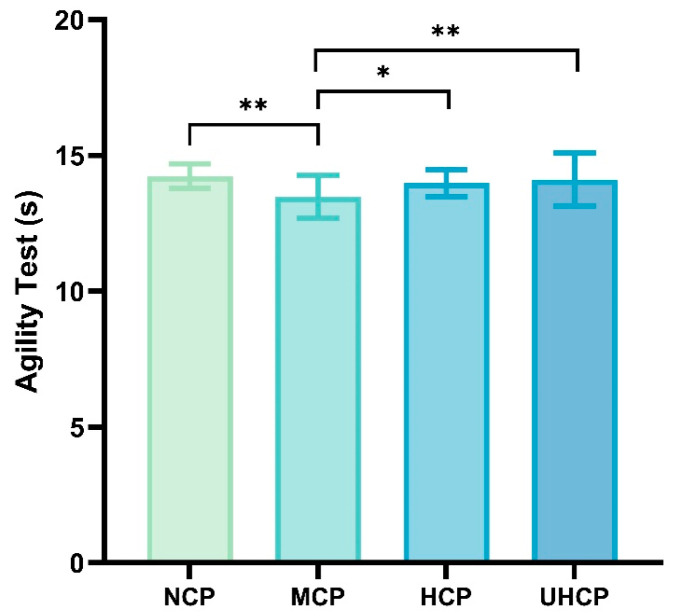
Agility performance under different compression pant pressure levels (* *p* < 0.05, ** *p* < 0.01).

**Figure 6 sensors-25-04875-f006:**
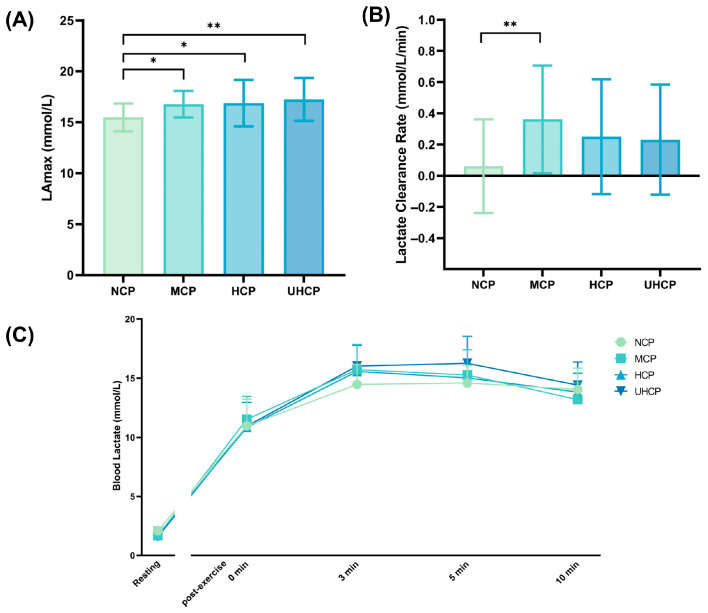
Post-exercise blood lactate responses under different compression pant pressure levels: (**A**) peak lactate concentration (LAmax), (**B**) lactate clearance rate at 10 min post-exercise, and (**C**) temporal changes in blood lactate levels (* *p* < 0.05, ** *p* < 0.01).

**Figure 7 sensors-25-04875-f007:**
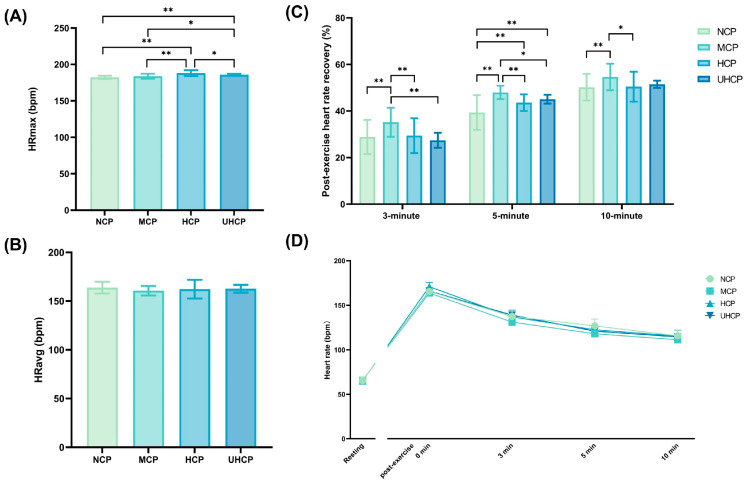
Heart rate responses under different compression pant pressure levels: (**A**) maximum heart rate (HRmax) during exercise, (**B**) average heart rate (HRavg), (**C**) heart rate recovery rates at 3, 5, and 10 min post-exercise, and (**D**) heart rate change trends before and after exercise (* *p* < 0.05, ** *p* < 0.01).

**Table 1 sensors-25-04875-t001:** Participant characteristics, expressed as x (mean) and SD (standard deviation); young adult males; *n* = 20.

	($\bar{x}$ ± SD)
Age (year)	19.9 ± 1.2
Height (cm)	177.5 ± 1.8
Body mass (kg)	65.4 ± 3.1
BMI ^1^ (kg/m^2^)	20.7 ± 1.1

^1^ BMI: body mass index.

**Table 2 sensors-25-04875-t002:** Results of 30 s Wingate test under different compression pant pressure levels.

	NCP	MCP	HCP	UHCP
BW/kg	4.90 ± 0.23			
TPP/ms	4906.18 ± 1817.55	3778.13 ± 1142.47 *	4637.49 ± 1455.34	4500.92 ± 1879.23
PP/(W/kg)	10.17 ± 0.29	10.30 ± 0.45	10.51 ± 0.44 *	10.34 ± 0.53
AP/(W/kg)	8.09 ± 0.16	8.72 ± 0.89 **	8.07 ± 0.26 ^##^	8.07 ± 0.19 ^##^
MP/(W/kg)	5.61 ± 0.21	5.56 ± 0.5	5.49 ± 0.31	5.66 ± 0.29
PD/(W/kg)	0.16 ± 0.02	0.14 ± 0.02 *	0.17 ± 0.03 ^##^	0.17 ± 0.04 ^#^
TEP/J	14,430.77 ± 631.59	15,038.57 ± 891.72 *	14,486.36 ± 691.59 ^#^	14,393.38 ± 971.91 ^#^

* *p* < 0.05, ** *p* < 0.01 vs. NCP; ^#^ *p* < 0.05, ^##^ *p* < 0.01 vs. MCP. BW: Brake Weight; TPP: Time at Peak Power; PP: Peak Power; AP: Average Power; MP: Minimum Power; PD: Power Drop; TEP: Total Energy Produced.

## Data Availability

The original contributions presented in this study are included in the article, and further inquiries can be directed to the corresponding author.

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
