# Peer review of "Effects of Compression Pants with Different Pressure Levels on Anaerobic Performance and Post-Exercise Physiological Recovery: Randomized Crossover Trial"

_sensors, 2025, doi:10.3390/s25154875_

Round 1

Reviewer 1 Report

Comments and Suggestions for Authors
  1. Regarding the textual content, what is the purpose of including the section from lines 87 to 93 in the "Participants and Methods" section of the article?

  2. The research background should be strengthened. The current discussion on the effects of compression garments on exercise in the Introduction section lacks recent references.

  3. In Section 2.3.1, pressure measurements were conducted on NCP, HCP, and UHCP, whereas in Section 3.1 (Results), pressure tests were reported for MCP, HCP, and UHCP. There is an inconsistency between these sections. The author is requested to verify and clarify this discrepancy.

  4. Three types of exercises were selected for the exercise tasks; however, the rationale behind choosing these specific exercises was not clearly explained. Additionally, the relationship between these selected exercises and the muscle groups chosen for pressure testing remains unclear.

  5. Post-exercise recovery, as part of the experimental protocol, requires more detailed description. For example, how long was heart rate data collected? What is the rationale for selecting rest intervals of 0, 3, 5, and 10 minutes for data acquisition?

  6. The Discussion section should be expanded to include future research directions and a discussion of the study’s limitations.

Comments on the Quality of English Language

The manuscript requires language refinement and professional editing to improve clarity and readability.

Author Response

Response to Reviewer 1 Comments

1. Summary

Thank you very much for taking the time and effort to review our manuscript. We truly appreciate your constructive comments and valuable suggestions. We have carefully revised the manuscript in response to the reviewers' feedback. Please find the detailed responses below, and the corresponding revisions have been clearly marked in red font in the re-submitted files for your convenience. We are grateful for your dedication and sincerely hope that the revised version meets your expectations. We respectfully invite you to review the manuscript again.

2.Point-by-point response to Comments and Suggestions for Authors

Comments 1: Regarding the textual content, what is the purpose of including the section from lines 87 to 93 in the "Participants and Methods" section of the article?

Response 1: Thank you very much for your valuable comment, and we sincerely appreciate your careful attention in pointing this out.

The section from lines 87 to 93 in the "Participants and Methods" was mistakenly included and does not contribute meaningfully to the manuscript. We have recognized this as a writing error and have removed these lines in the revised version.

Comments 2: The research background should be strengthened. The current discussion on the effects of compression garments on exercise in the Introduction section lacks recent references.

Response 2: Thank you for your valuable comment.

We agree that the research background needed to be strengthened and have revised the Introduction section accordingly. Specifically, we have added several recent references from the past five years to enhance the discussion on the effects of compression garments on exercise. These updates can be found in lines 61–82, 86–98, 102–107, and 118–126 of the revised manuscript. These additions aim to provide a more comprehensive and up-to-date overview of recent findings in this field and to better support the rationale for our study.

Comments 3: In Section 2.3.1, pressure measurements were conducted on NCP, HCP, and UHCP, whereas in Section 3.1 (Results), pressure tests were reported for MCP, HCP, and UHCP. There is an inconsistency between these sections. The author is requested to verify and clarify this discrepancy.

Response 3: Thank you very much for pointing out this inconsistency.

Upon careful review, we found that the term “NCP” in Section 2.3.1 was a writing error. It should have been “MCP” to match the pressure test groups reported in Section 3.1 (Results), namely MCP, HCP, and UHCP. This correction has been made at Line 206 in the revised manuscript to ensure consistency between the methods and results descriptions.

Comments 4: Three types of exercises were selected for the exercise tasks; however, the rationale behind choosing these specific exercises was not clearly explained. Additionally, the relationship between these selected exercises and the muscle groups chosen for pressure testing remains unclear.

Response 4: Thank you very much for your valuable comment.

To address the issue of insufficient explanation regarding the rationale for selecting the three exercise tasks and their relationship with the muscle groups chosen for pressure measurement, we have added further clarifications in the revised manuscript at the following locations:

In Section 3.1 (Lines 278–289), we explain the rationale for selecting the four muscle groups (rectus femoris, vastus lateralis, gastrocnemius, and tibialis anterior), highlighting their respective roles in anaerobic performance and their relevance for evaluating the effects of compression garments.

In Section 3.2 (Lines 313–319), we describe the rationale for selecting the vertical jump test and clarify its relevance to the measured muscle groups, particularly the quadriceps (rectus femoris and vastus lateralis) and gastrocnemius, which are functionally involved in explosive lower-limb power output.

In Section 3.3 (Lines 329–337), we provide the rationale for using the Illinois Agility Test, emphasizing its demands on neuromuscular coordination, rapid change of direction, and the engagement of tibialis anterior, gastrocnemius, and quadriceps in supporting dynamic lower-limb control.

In Section 3.4 (Lines 346–353), we explain the rationale for selecting the 30-second Wingate test as a gold standard for anaerobic performance assessment, and its strong association with high-intensity activation of the quadriceps muscles.

We hope these additions address your comment. Thank you again for your constructive feedback, which has helped us improve the clarity and completeness of the manuscript.

Comments 5: Post-exercise recovery, as part of the experimental protocol, requires more detailed description. For example, how long was heart rate data collected? What is the rationale for selecting rest intervals of 0, 3, 5, and 10 minutes for data acquisition?

Response 5: Thank you very much for your helpful comment.

To address this issue and make the rationale for heart rate data collection more explicit, we have provided additional clarification in the revised manuscript. Specifically, the rationale for selecting the 0, 3, 5, and 10-minute time points has been added in Section 2.3.6 (Lines 251–256 and 257–259). These time points correspond to immediate, early, and late phases of post-exercise recovery and are commonly adopted in previous studies evaluating autonomic function after high-intensity anaerobic exercise.

In addition, we have cited relevant literature (e.g., Kurniawan et al., 2022; Michael et al., 2017; Pierpont et al., 2000) to support the time-point selection and strengthen the methodological justification.

Comments 6: The Discussion section should be expanded to include future research directions and a discussion of the study’s limitations.

Response 6: Thank you very much for your constructive comment.

To address this issue, we have expanded the Discussion section by adding a new paragraph at the end of Section 4 (Lines 527–548) to discuss the limitations of the current study and provide directions for future research. Specifically, we have acknowledged the limited generalizability of the findings due to the homogeneous participant sample (young males with similar anthropometric characteristics), the inability to measure dynamic compression pressure during exercise because of sensor limitations, and the absence of additional physiological indicators such as electromyography and muscle oxygenation. Furthermore, we highlighted the need to explore the long-term effects of compression garments during repeated anaerobic training sessions and suggested including more diverse populations and advanced monitoring techniques in future studies.

3.Response to Comments on the Quality of English Language

Point 1: The English could be improved to more clearly express the research.

Response 1:  Thank you for your comment.

To address this issue, we have thoroughly revised the manuscript to improve the clarity, readability, and overall language quality. The text has been carefully reviewed and edited by a native English speaker with experience in academic writing. We believe the revised version now more clearly and accurately conveys the research content.

Reviewer 2 Report

Comments and Suggestions for Authors

This manuscript presents a well-designed randomized crossover trial investigating the effects of compression pants with varying pressure levels on anaerobic performance and post-exercise recovery in healthy male university students. The study addresses an important gap in the literature regarding the differential effects of compression garment pressure levels on anaerobic performance metrics. While the study is generally well-conducted and reported, several methodological and reporting issues need to be addressed before the manuscript can be considered for publication.

  1. The manuscript does not provide clear criteria for how the different pressure levels (moderate, high, ultra-high) were classified or determined. Are these classifications based on manufacturer specifications or measured values? Please clarify.Moreover, the pressure measurements may change during exercise. Provide more details about the specific brands/models of compression pants used.
  2. For crossover designs, it's important to include period and carryover effects in the analysis. Please confirm whether these were accounted for in the statistical model.The inverted-U hypothesis is interesting but somewhat speculative based on the current data. Consider tempering this conclusion or providing more direct evidence.
  3. Sample size (n=20) meets a priori calculations but may lack power for subgroup analyses (e.g., inter-individual responses).
  4. The discussion should address why MCP showed advantages in some metrics but not others. Is there a theoretical framework that could explain these differential effects?
  5. Participants were narrowly selected (males, 175–180 cm, 60–70 kg). Generalizability to females, athletes, or varied anthropometrics is unclear.This limitation and recommend future studies with diverse cohorts should be discussed.
  6. The literature review could be strengthened by more recent references (many are from 2010-2014). Several relevant 2020-2024 publications on compression garments should be incorporated.
  7. The content in Line 87-93 is possibly the writing mistake and should be deleted.

Author Response

Response to Reviewer 2 Comments

1.Summary

We sincerely thank the reviewer for their thorough and constructive evaluation of our manuscript. We greatly appreciate the time and effort invested in providing detailed feedback, which has helped us identify important areas for improvement. We have carefully considered all comments and suggestions, and have made corresponding revisions to the manuscript to enhance its clarity, rigor, and overall quality. Point-by-point responses to each comment are provided below, with all changes clearly marked in the revised files using red font.

2.Point-by-point response to Comments and Suggestions for Authors

Comments 1: The manuscript does not provide clear criteria for how the different pressure levels (moderate, high, ultra-high) were classified or determined. Are these classifications based on manufacturer specifications or measured values? Please clarify. Moreover, the pressure measurements may change during exercise. Provide more details about the specific brands/models of compression pants used.

Response 1: Thank you very much for your insightful comments. We have addressed your suggestions as follows:

1.Classification Criteria for Compression Levels:

The classification of garments into Moderate Compression Pressure (MCP), High Compression Pressure (HCP), and Ultra-High Compression Pressure (UHCP) was based on measured interface pressure values at rest, not on manufacturer claims. To justify this classification, we have added supporting references in the revised manuscript (see Line 207), following established guidelines in compression therapy literature.

2.Brand and Manufacturer Information:

We appreciate your suggestion and have now included the full brand and manufacturer details of the three compression tights used in this study in Lines 190–192 of the revised manuscript, to enhance clarity and transparency.

3.Pressure Measurement During Exercise:

We acknowledge that interface pressure may vary during exercise due to body movement and fabric displacement. However, multiple previous studies have adopted static-state pressure measurements to ensure the consistency and reproducibility of compression quantification across participants and studies. According to MacRae et al. (2011), static pressure measurements exhibit a low error rate (less than 5%), whereas dynamic measurements are prone to motion artifacts with error rates ranging from 15% to 28%. Based on these considerations, we employed static measurements at rest to maintain methodological consistency. This limitation has also been discussed in the revised manuscript (see Lines 533–538), where we suggest that future research should incorporate real-time, motion-compatible pressure monitoring to capture dynamic changes more accurately.

Comments 2: For crossover designs, it's important to include period and carryover effects in the analysis. Please confirm whether these were accounted for in the statistical model. The inverted-U hypothesis is interesting but somewhat speculative based on the current data. Consider tempering this conclusion or providing more direct evidence.

Response 2: Thank you very much for your valuable comments.

To address your first concern regarding the crossover design, we have revised the Statistical Analysis section of the manuscript (see Lines 270–273) to clarify that period and sequence (carryover) effects were examined using a linear mixed-effects model. Specifically, “condition” and “period” were included as fixed effects, while “subject” was modeled as a random effect. The analysis revealed no significant carryover effects, thereby confirming that the crossover design was properly accounted for in the statistical model.

Regarding the second point on the "inverted-U hypothesis," we agree that this interpretation may appear speculative based on the current dataset. To address this, we have revised the relevant content in the Discussion section (Lines 423–426 and 452–454), reframing the hypothesis in a more cautious and data-aligned manner. We now describe the relationship between compression intensity and performance outcomes as nonlinear and task-specific, rather than suggesting a definitive dose–response curve. This revision more accurately reflects the observed variability across different exercise modalities. Additionally, as noted in Lines 538–541, we acknowledge that further research incorporating mechanistic measurements (e.g., intramuscular pressure, neuromuscular activation, muscle oxygenation) is needed to better understand and validate these regulatory patterns.

Comments 3: Sample size (n=20) meets a priori calculations but may lack power for subgroup analyses (e.g., inter-individual responses).

Response 3: Thank you very much for your thoughtful comment.

We agree that while the current sample size (n = 20) meets the minimum requirement based on a priori power analysis for detecting within-subject main effects (effect size f = 0.3, α = 0.05, power = 0.80), it may indeed limit the statistical power to detect more nuanced effects, such as subgroup differences or inter-individual variability. Our primary objective in this study was to evaluate the overall effects of different compression pressure levels using a repeated-measures crossover design, where each participant served as their own control, thereby reducing between-subject variability and improving statistical efficiency.

However, we acknowledge that this sample size is not sufficient to support robust subgroup analyses (e.g., by training level, compression responsiveness), which typically require larger cohorts to detect interaction or stratified effects. This limitation has now been clearly noted in the revised Discussion section (Lines 544–548). We also suggest that future studies should consider recruiting larger and more heterogeneous populations to further explore individual response variability and identify potential responder subtypes.

Comments 4: The discussion should address why MCP showed advantages in some metrics but not others. Is there a theoretical framework that could explain these differential effects?

Response 4: We sincerely thank the reviewer for this insightful comment.

We agree that the differential effects of moderate compression pressure (MCP) across various performance metrics warrant further theoretical explanation. To address this, we have expanded the discussion in Lines 458–470 by incorporating the sensorimotor integration framework, which emphasizes the dynamic interaction between afferent sensory input and motor planning. This framework helps explain why MCP demonstrated superior outcomes in agility-based and sustained-effort tasks, despite comparatively smaller gains in explosive output. The revised text provides a more nuanced and mechanistic interpretation of how compression garments may differentially influence task performance depending on pressure level and movement demands.

Comments 5: Participants were narrowly selected (males, 175–180 cm, 60–70 kg). Generalizability to females, athletes, or varied anthropometrics is unclear. This limitation and recommend future studies with diverse cohorts should be discussed.

Response 5: Thank you for this important comment.

We fully agree that the narrow inclusion criteria in our study may limit the generalizability of the findings. To address this issue, we have explicitly discussed this limitation in the revised Discussion section (Lines 528–533). Specifically, we acknowledge that participants were narrowly selected (healthy young males with similar anthropometric characteristics: 175–180 cm in height, 60–70 kg in body mass), which may restrict the applicability of the results to broader populations such as females, older adults, elite athletes, or individuals with different body types. We now recommend that future studies recruit more diverse cohorts to improve the external validity of the findings.

Additionally, we have clarified the rationale for our participant selection in the Introduction (Lines 127–136). As noted, this population—recreationally active males—is underrepresented in current compression garment literature despite being a high-frequency user group, particularly in anaerobic training settings such as resistance training and team sports. We therefore considered it both practically relevant and scientifically meaningful to focus on this demographic in the present study.

Comments 6: The literature review could be strengthened by more recent references (many are from 2010-2014). Several relevant 2020-2024 publications on compression garments should be incorporated.

Response 6: Thank you for your helpful suggestion.

To strengthen the literature review and address this issue, we have incorporated several recent studies published between 2020 and 2024 on compression garments. These updated references have been added to the Introduction section, specifically in lines 61–82, 86–98, 102–107, and 118–126 of the revised manuscript, to provide a more current and comprehensive overview of the field.

Comments 7: The content in Line 87-93 is possibly the writing mistake and should be deleted.

Response 7: Thank you for pointing out this detail.

We have carefully reviewed the content in lines 87–93 and confirmed that it was a writing error, as it does not contribute meaningfully to the manuscript. The corresponding text has now been deleted from the revised manuscript.

3.Response to Comments on the Quality of Figures

Point 1: Figures and tables can be improved

Response 1: We sincerely thank the reviewer for this helpful suggestion.

In response, we have improved the overall quality and clarity of the figures and tables. Additionally, we have added a new figure (Figure 2) to illustrate the compression garment pressure testing procedure and the sensor placement on key lower limb muscles. We hope these revisions enhance the visual presentation and effectively support the interpretation of the results.

Round 2

Reviewer 1 Report

Comments and Suggestions for Authors

It has been modified and is suitable for publication in journals.

Reviewer 2 Report

Comments and Suggestions for Authors

The authors have addressed my main concerns about this work.